# Pregnancy duration and breast cancer risk

Anders Husby [1,2], Jan Wohlfahrt[1], Nina Øyen[1,3,4] & Mads Melbye [1,5,6]

Full-term pregnancies reduce a woman's long-term breast cancer risk, while abortions have been shown to have no effect. The precise minimal duration of pregnancy necessary to lower a woman's breast cancer risk is, however, unknown. Here we provide evidence which point to the protective effect of pregnancy on breast cancer risk arising precisely at the 34th pregnancy week. Using a cohort of 2.3 million Danish women, we found the reduction in breast cancer risk was not observed for pregnancies lasting 33 weeks or less, but restricted to those pregnancies lasting 34 weeks or longer. We further found that parity, socioeconomic status, and vital status of the child at birth did not explain the association, and also replicated our finding in data from 1.6 million women in Norway. We suggest that a distinct biological effect introduced around week 34 of pregnancy holds the key to understand pregnancy-associated breast cancer protection.

[1] Department of Epidemiology Research, Statens Serum Institut, DK-2300 Copenhagen, Denmark. [2] Department of Biomedical Data Science, Stanford University School of Medicine, Stanford, CA 94305, USA. [3] Department of Global Public Health and Primary Care, Faculty of Medicine, University of Bergen, N-5020 Bergen, Norway. [4] Department of Medical Genetics, Haukeland University Hospital, N-5021 Bergen, Norway. [5] Department of Clinical Medicine, University of Copenhagen, DK-2100 Copenhagen, Denmark. [6] Department of Medicine, Stanford University School of Medicine, Stanford, CA 94305, USA. Correspondence and requests for materials should be addressed to M.M. (email: mme@ssi.dk)

Breast cancer is the most common malignant cancer in women and a major cause of disease burden worldwide[1]. Both the number and timing of a woman's childbirths have long been known to influence her breast cancer risk[2], but how these factors influence breast cancer etiology is not well understood. Previously, full-term pregnancies in early life (<30 years) have consistently been associated with a long-term reduced risk of breast cancer[3,4]. Conversely, a transient increased breast cancer risk immediately following full-term pregnancies have been observed[5]. Induced abortions and other pregnancies of short duration have, on the other hand, been shown not to influence breast cancer risk[6,7]. We hypothesized that by investigating pregnancies of intermediate to long duration in early life (including stillbirths, preterm, and term livebirths) we could determine the minimal duration of pregnancy associated with a reduced risk of breast cancer and thereby potentially point to underlying mechanisms of the protective effect.

Taking advantage of the Danish national registries on childbirths and cancer, we established a nationwide cohort including 2.3 million women with detailed information on reproductive history from 1978 to 2014, and assessed the association between the duration of a pregnancy and the long-term risk of breast cancer. We replicated this analysis in an equivalent Norwegian cohort of 1.6 million women.

## Results

**Cohort description and age at pregnancy**. The Danish cohort consisted of 2,311,332 women, with altogether 3,275,559 childbirths. The women were followed for 46,128,328 person-years (average 20.0 years follow-up per woman) and 61,349 (2.7%) developed breast cancer. We focused on follow-up from 10 years or more after pregnancy, to highlight the long-term effect of pregnancy on breast cancer risk. Table 1 shows number of breast cancer events and follow-up time according to number of childbirths, age at first childbirth, and duration of latest pregnancy.

Figure 1 shows the long-term relative risk (RR) of breast cancer after first childbirth by age at delivery, adjusted for different socioeconomic variables. Overall, we found a first childbirth before 30 years of age to decrease the long-term breast cancer risk. Further, to investigate the effect of both first and subsequent childbirths in early age on long-term breast cancer risk, we estimated the effect of first, second, and third childbirth, compared with one childbirth less (Supplementary Fig. 1). For childbirths before 30 years of age, women's long-term breast cancer risk was reduced for the first childbirth (on average 5.0% (95% CI: 2.1% to 7.8%)), the second (on average 6.4% (95% CI: 3.9% to 8.8%)), and the third childbirth (on average 9.4% (95% CI: 6.4% to 12.2%)). For childbirths at 30 years or later, we did not observe a consistent, overall reduced long-term breast cancer risk (first birth: −8.7% (95% CI: −12.8% to −4.8%), second birth: 3.4% (95% CI: 0.7% to 6.0%), third birth: 5.3% (95% CI: 2.7% to 7.8%)).

**Pregnancy duration and breast cancer risk**. We speculated whether the observed reduced long-term breast cancer risk following any birth at an early age varied by pregnancy duration. To study this, we included information on pregnancy duration. Figure 2a shows the long-term RR of breast cancer after an early age childbirth compared with one childbirth less, by pregnancy duration. We noted a distinctive difference in the cancer risk associated with pregnancies lasting 34 weeks and longer compared with pregnancies lasting 33 weeks or less. Whereas pregnancies 33 weeks or less were not associated with long-term breast cancer risk (on average 2.3% (95% CI: −10.0% to 13.2%)

risk reduction per birth), pregnancies lasting 34 weeks or longer were associated with a substantially reduced risk (on average 12.9% (95% CI: 11.4% to 14.3%) risk reduction per birth). Additionally, to examine the role of breastfeeding, we investigated the effect of stillbirths, which are not breastfeed, and found that both live births and stillbirths were associated with reduced breast cancer risk, but only if delivered at week 34 or later (Table 2).

We replicated the analyses in a similar cohort of 1,635,839 Norwegian women with altogether 2,420,518 pregnancies identified in the National Registry and the Norwegian Medical Birth Registry. The women were followed for 35,171,205 person-years, in which 24,095 developed invasive breast cancer.

Figure 2b illustrates the long-term RR of breast cancer after an early age childbirth compared with one childbirth less, according to duration of pregnancy, in the Norwegian cohort. As shown, we found a pattern identical to the results obtained in the Danish cohort. In the Norwegian cohort, the average reduction in long-term breast cancer risk associated with a pregnancy lasting 33 weeks or less were 2.9% (95% CI: −7.7% to 12.6%), whereas the reduction with pregnancies lasting 34 weeks or more were 14.5% (95% CI: 13.1% to 15.8%).

When we combined the Danish and Norwegian cohorts (Fig. 2c), the reduction in long-term breast cancer risk associated with early childbirth was 2.4% (95% CI: −5.6% to 9.7%) for pregnancies lasting 33 weeks or less and 13.6% (95% CI: 12.6% to 14.5%) for pregnancies lasting 34 weeks or longer. The reduced risk of breast cancer for pregnancies lasting 34 weeks or more could have been modified by the number of previous pregnancies lasting 33 weeks or less, but the risk reduction was similar for no previous births <34 weeks, 13.5% (95% CI: 12.5% to 14.5%); one previous birth <34 weeks, 16.9% (95% CI: 10.2% to 23.1%); or two or more previous births <34 weeks, 37.7% (95% CI: 7.5% to 58.1%). Furthermore, to avoid a possible distinct effect of a woman's first childbirth on cancer risk, we focused on the effect of a second, third, or any additional childbirth, among women in Denmark and Norway (Supplementary Fig. 2), and found that the reduction in long-term breast cancer risk associated with early age childbirth was 1.2% (95% CI: −11.2% to 12.4%) for pregnancies lasting 33 weeks or less and 16.3% (95% CI: 14.9% to 17.8%) for pregnancies lasting 34 weeks or longer.

We performed additional sensitivity analyses to evaluate the association between a specific duration of a pregnancy and a woman's long-term breast cancer risk (see Supplementary Fig. 3 for effect of relative birthweight and Supplementary Fig. 4 for effect of induced abortions and childbirths). Adjusting for individual-level socioeconomic differences, we found no strong confounding effect of socioeconomic factors on breast cancer risk in the analysis of pregnancy duration (Supplementary Fig. 5). In analysis of threshold models, where all risk reduction occur in pregnancies lasting longer than a specific duration, we furthermore found the best fit of data for a threshold of 34 weeks duration of pregnancy (Supplementary Fig. 6A). The same conclusion was reached when allowing the protective effect in the models to vary by parity and country (Supplementary Fig. 6B and Supplementary Fig. 6C).

## Discussion

The strongest known modifier of a woman's breast cancer risk is her reproductive history. Thus, early age full-term pregnancies and an increasing number of childbirths[3,4] result in a lowered breast cancer risk, whereas abortions do not influence breast cancer risk[6,7]. Previous studies on preterm birth and breast cancer risk have nevertheless not had statistical power to show any specific effect of pregnancy duration on breast cancer risk[8–11]. In the present study, we provide evidence that the protection

**Table 1 Breast cancer events and person-years according to number of childbirths, age at first childbirth, and duration of latest pregnancy in the Danish and the Norwegian cohort[a]**

| Cohort characteristic | The Danish cohort | | The Norwegian cohort | |
|---|---|---|---|---|
| | Breast cancer events (%) | Persons-years in 1000s (%) | Breast cancer events (%) | Persons-years in 1000s (%) |
| *Number of childbirths* | | | | |
| 0 | 8028 (13.1) | 23,370 (50.7) | 2880 (13.9) | 13,861 (55.1) |
| 1 | 10,523 (17.1) | 4418 (9.6) | 2996 (14.4) | 1850 (7.3) |
| 2 | 28,046 (45.7) | 11,938 (25.9) | 8582 (41.3) | 5268 (20.9) |
| 3 | 11,468 (18.7) | 4888 (10.6) | 4639 (22.3) | 2965 (11.8) |
| 4 | 2667 (4.3) | 1175 (2.5) | 1313 (6.3) | 901 (3.6) |
| ≥5 | 617 (1.0) | 340 (0.7) | 367 (1.8) | 324 (1.3) |
| *Age at first childbirth (years)* | | | | |
| Nulliparous | 8028 (13.1) | 23,370 (50.7) | 2880 (13.9) | 13,861 (55.1) |
| <20 | 8538 (13.9) | 4109 (8.9) | 3021 (14.5) | 2490 (9.9) |
| 20–21 | 9630 (15.7) | 4535 (9.8) | 3692 (17.8) | 2682 (10.7) |
| 22–23 | 10,150 (16.5) | 4569 (9.9) | 3320 (16.0) | 2221 (8.8) |
| 24–25 | 8956 (14.6) | 3814 (8.3) | 2957 (14.2) | 1695 (6.7) |
| 26–27 | 6373 (10.4) | 2531 (5.5) | 1979 (9.5) | 1018 (4.0) |
| 28–29 | 4087 (6.7) | 1489 (3.2) | 1247 (6.0) | 579 (2.3) |
| ≥30 | 5587 (9.1) | 1712 (3.7) | 1681 (8.1) | 621 (2.5) |
| *Duration of latest pregnancy (weeks)[b]* | | | | |
| Nulliparous | 8028 (13.1) | 23,370 (50.7) | 2880 (13.9) | 13,861 (55.1) |
| 20–27 | 34 (0.1) | 12 (0.0) | 29 (0.1) | 16 (0.1) |
| 28–29 | 38 (0.1) | 18 (0.0) | 22 (0.1) | 15 (0.1) |
| 30 | 33 (0.1) | 14 (0.0) | 20 (0.1) | 13 (0.1) |
| 31 | 33 (0.1) | 16 (0.0) | 40 (0.2) | 18 (0.1) |
| 32 | 54 (0.1) | 26 (0.1) | 44 (0.2) | 25 (0.1) |
| 33 | 78 (0.1) | 35 (0.1) | 62 (0.3) | 36 (0.1) |
| 34 | 101 (0.2) | 54 (0.1) | 92 (0.4) | 61 (0.2) |
| 35 | 178 (0.3) | 81 (0.2) | 154 (0.7) | 100 (0.4) |
| 36 | 360 (0.6) | 163 (0.4) | 262 (1.3) | 170 (0.7) |
| 37 | 719 (1.2) | 332 (0.7) | 507 (2.4) | 334 (1.3) |
| 38 | 1626 (2.7) | 798 (1.7) | 1254 (6.0) | 792 (3.1) |
| 39 | 3091 (5.0) | 1519 (3.3) | 2959 (14.2) | 1795 (7.1) |
| 40 | 6369 (10.4) | 3062 (6.6) | 3902 (18.8) | 2523 (10.0) |
| 41 | 2846 (4.6) | 1446 (3.1) | 2877 (13.8) | 1886 (7.5) |
| ≥42 | 1288 (2.1) | 685 (1.5) | 1777 (8.6) | 1222 (4.9) |
| Missing duration of pregnancy in birth register | 1493 (2.4) | 662 (1.4) | 638 (3.1) | 471 (1.9) |
| Childbirths registered in civil registers[c] | 34,980 (57.0) | 13,835 (30.0) | 3217 (15.5) | 1799 (7.1) |

[a]All events and person-years from 10 years after latest childbirth
[b]Pregnancies registered to have lasted less than 20 weeks or more than 45 gestational weeks were also included in the analysis as separate categories (see Statistical analyses), but constituted combined only <0.01% and 0.13% of observation time in Denmark and Norway, respectively
[c]Childbirths registered in the civil registration systems, but not in the Birth Registers. Predominantly childbirths before January 1, 1978 in Denmark and January 1, 1967 in Norway. After these dates only 3.34% and 3.85% of childbirths are not reported in the Medical Births Registers, in Denmark and Norway, respectively

introduced by a pregnancy takes place around a specific pregnancy week. Using Danish nationwide registers, we found the minimal pregnancy length associated with a substantial reduced risk of long-term breast cancer to be 34 weeks, whereas a pregnancy length of 33 weeks or less did not confer a reduction in risk. The exact same result was obtained in a Norwegian replication cohort based on similar nationwide register data.

It has been hypothesized that a woman's first pregnancy has a special influence on mammary tissue structural remodeling[12], and that this might explain the reduced risk of breast cancer later in life. It has specifically been suggested that pregnancy-induced differentiation of breast cells at this first pregnancy might make them less sensitive to influences from external carcinogenic stimuli[13]. However, we found that additional childbirths further reduce breast cancer risk and that the effect observed by these additional births is at a similar level as observed for the first birth. In addition, the specific effect of a pregnancy lasting 34 weeks or longer on later breast cancer risk was also evident in subsequent

pregnancies. Taken together this gives little support to the hypothesis of a decisive and particularly distinct effect on mammary tissue caused by the first pregnancy that associates with later breast cancer risk.

Multiple studies have pointed to persistent changes in gene expression[14,15], epigenetic structure[16–20], and epithelial stem cell composition[18,21] in the mammary gland following pregnancy. However, the mechanisms proposed for pregnancy-induced breast cancer protection have neither been substantiated or replicated. Our novel finding that pregnancy-induced breast cancer protection is obtained within a narrow time window, late in pregnancy, will enable a meticulous investigation of the causal factor behind this striking effect. Furthermore, a precise characterization of the factor responsible for the effect will be helped by our observation that each early age pregnancy offers cumulative protection against breast cancer. Altogether, our results can open a path to explore the specific biological mechanism behind the impact of pregnancies on subsequent breast cancer risk.

Our observations are in line with findings from mammalian breast cancer models that show a protective effect of pregnancy introduced close to term[22]. In theory, late-pregnancy stimuli that transform the breast tissue to a stage represented by a lowered breast cancer risk could originate from both the mother and the fetus. However, findings by us and others provide little evidence for a fetal involvement since infant sex[23], infant absolute birthweight[10,24] or, as we show, fetal growth restriction and vital status of the infant at birth do not influence the long-term breast cancer risk.

Breastfeeding, and in particular the total breastfeeding duration, has been proposed to protect against breast cancer[25] and could potentially explain the association with pregnancy length. However, at least two of our findings strongly argue against this being the case. First, we found an equivalent protective effect on breast cancer risk of stillbirths and live births from the 34th gestational week. Second, the pregnancy-induced risk reduction of breast cancer was restricted to young women below 30 years of age at childbirth, whereas the total breastfeeding duration is shortest for mothers younger than 30 years[26] and does not vary markedly by gestational length of pregnancy[27].

High levels of alcohol consumption are found to be associated with an increased risk of breast cancer[28,29], and there are reports of an association between heavy alcohol consumption and pre-term birth[30,31], why alcohol consumption could be a potential factor in the association between pregnancy duration and breast cancer risk. There are however large differences in alcohol consumption between Denmark and Norway, with studies of drinking patterns finding that this is the case both with regards to drinking frequency and volume[32,33], with differences in alcohol consumption being especially pronounced during pregnancy[30,34]. Given the marked differences in alcohol consumption between Denmark and Norway, and the identical findings on the association between pregnancy duration and breast cancer risk, we find it unlikely that alcohol consumption serves as a major

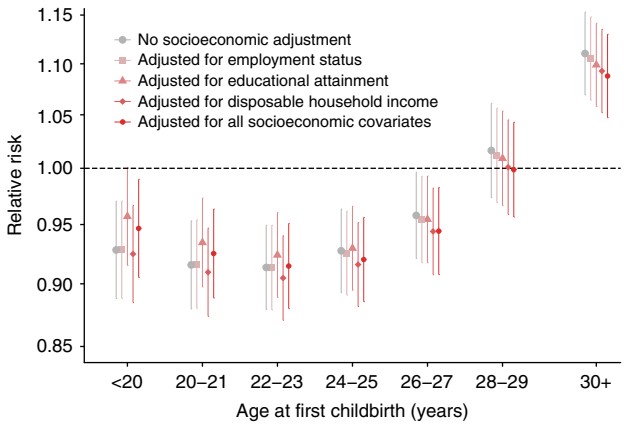

**Fig. 1** Effect of different socioeconomic factors on long-term relative risk of breast cancer after first childbirth in Denmark compared with nulliparous, by age at delivery. Error bars indicate 95% confidence intervals

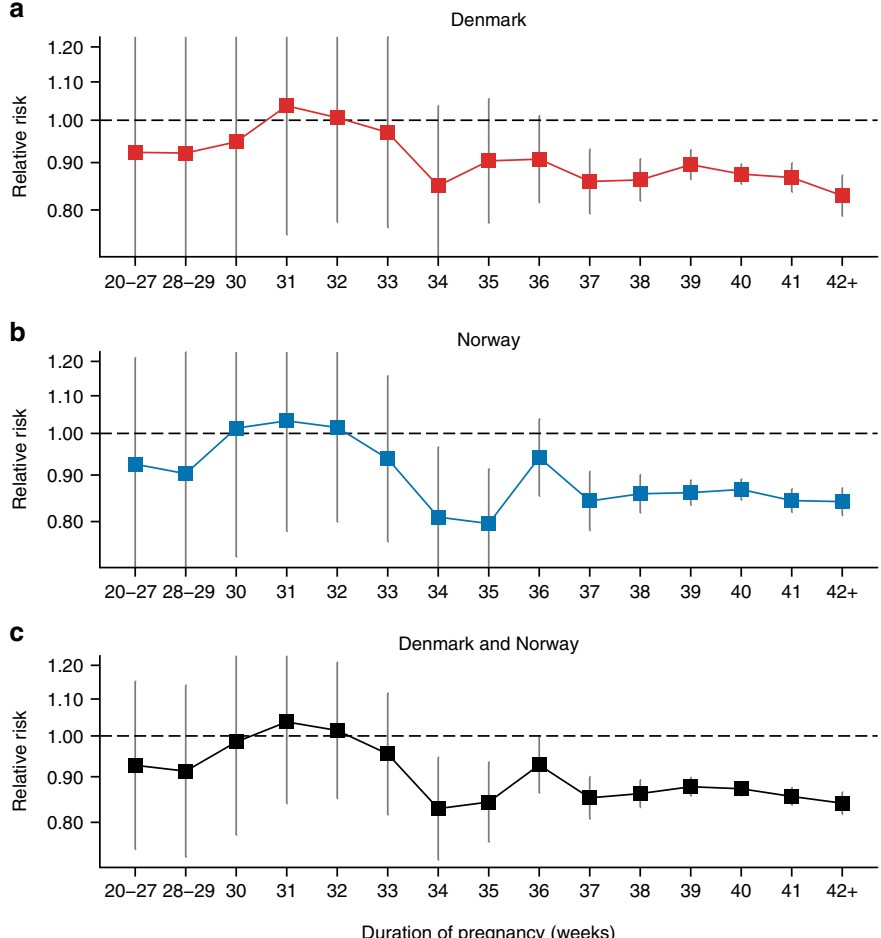

**Fig. 2** Long-term relative risk of breast cancer after an early age childbirth compared with one childbirth less, according to duration of pregnancy. **a** Denmark, **b** Norway, and **c** combined. Error bars indicate 95% confidence intervals

**Table 2 Long-term relative risk of breast cancer after an early age childbirth compared with one childbirth less, according to the duration of pregnancy and type of childbirth in the Danish cohort**

| Type of childbirth[a] | Duration of Pregnancy | |
|---|---|---|
| | <34 weeks | ≥34 weeks |
| *Unadjusted for socioeconomic factors* | | |
| Live birth | 0.99 (0.87–1.12) | 0.87 (0.86–0.89) |
| Stillbirth | 0.84 (0.54–1.30) | 0.69 (0.51–0.94) |
| *Adjusted for socioeconomic factors*[b] | | |
| Live birth | 0.99 (0.87–1.12) | 0.87 (0.85–0.88) |
| Stillbirth | 0.84 (0.54–1.30) | 0.69 (0.51–0.94) |

Unadjusted and adjusted for socioeconomic factors (with corresponding 95% confidence intervals)
[a]Of the total number of childbirths in the Danish cohort with known duration of pregnancy 3442 (0.18%) were stillbirths before week 34, 5970 (0.31%) were stillbirths at week 34 or later, 30,437 (1.56%) were live births before week 34 and 1,912,529 (97.96%) were live births at week 34 or later
[b]Adjustment for disposable household income, level of educational attainment, and employment status

confounding factor for the association between pregnancy duration and maternal breast cancer risk.

Using prospective national registers ensured high validity, negligible selection bias, and minimal misclassification of women's number of childbirths, and timing of these births. In addition, the gestational duration of pregnancy is determined by medical professionals at the time of pregnancy which ensures proper classification of the pregnancy duration. The long follow-up and the nationwide scope of the study furthermore provided high statistical power, and the replication using Norwegian national registers underlined the validity of the findings. Finally, socioeconomic status and other potential confounding factors did not explain the association of minimal pregnancy duration and long-term breast cancer risk.

In conclusion, we found that each pregnancy in early age, and not only the first, is associated with a significant long-term protective effect against breast cancer. Furthermore, only pregnancies lasting 34 weeks or longer were associated with a reduction in breast cancer risk. The reduction in breast cancer risk was present regardless of whether the pregnancy ended in stillbirth or live birth, and therefore cannot be explained by breastfeeding. This suggests that a specific biological effect operating around week 34 of pregnancy induces long-term breast cancer protection.

## Methods

**Population registries**. We established a population-based cohort of Danish women by linking data from the Civil Registration System (CRS) with data from the Medical Birth Registry and the Danish Cancer Registry. The CRS contains detailed demographic information on all Danish residents, including linkage of women to their children's dates of birth. Since April 1, 1968, all Danish residents who were alive or born thereafter have been assigned a unique identification number in the CRS. This number permits information from different national registries to be linked together. All live and stillbirths in Denmark, with dates of birth, have been registered since 1973 in the Medical Birth Registry. Since 1978, gestational week at time of birth has been recorded. For sensitivity analyses, we obtained information on induced abortions in Denmark from the National Registry of Induced Abortions, where induced abortions have been mandatory reported to since 1939.

Information on breast cancer diagnoses was retrieved from the Danish Cancer Registry, which contains information on all cancers diagnosed in Denmark since 1943 and is considered close to complete[35]. From Statistics Denmark we acquired time-varying, individual-level socioeconomic data to address covariates potentially associated with reproduction and breast cancer[36]; educational attainment (since 1970), employment status (since 1976), and disposable household income (since 1990).

In Norway, we linked data from the National Registry, the Medical Birth Registry of Norway, and the Cancer Registry of Norway. The Medical Birth Registry has registered all births (including gestational week of delivery) since

1967[37] and the Cancer Registry is considered accurate and close to complete with regard to cancer diagnoses from 1953[38].

The research project was approved by institutional review for inclusion on Statens Serum Institutes permit for research projects given by the Danish Data Protection Agency (permit No. 2015-57-0102) and approved by the Regional Ethics Committee of Western Norway (permit 252.06).

**Subjects**. We established a cohort of all Danish women born between January 1, 1935 and December 31, 2002. Using the CRS number, we linked information on each woman's childbirths with the corresponding pregnancy duration (gestational week of delivery), and information on whether she developed invasive breast cancer. We furthermore established a cohort of all Norwegian women born between January 1, 1935 and December 31, 1994, with equivalent information on reproductive history and breast cancer.

**Statistical analyses**. Incidence rate ratios (in the following termed RR) of breast cancer by pregnancy history were estimated by log-linear Poisson regression in the Danish cohort, the Norwegian cohort, and the combined cohort. In Denmark, each woman was followed from January 1, 1978, or from her 12th birthday, whichever came later, until breast cancer, death, emigration or December 31, 2014, whichever came first. In Norway, each woman was followed from January 1, 1967, or from her 12th birthday, whichever came later, until breast cancer, death, emigration or December 31, 2006, whichever came first. All analyses were adjusted for effects of current age and time period in 5-year categories.

Pregnancy history was modeled by time-dependent variables as described previously[4]. Thus, instead of describing history by the total number of childbirths (i.e., RR of cancer in women with 1, 2, 3, or 4 births compared with women with 0 births), pregnancy history was evaluated by the RR for women with $n$ births compared with women with $n-1$ births (i.e., RR of cancer for 1 birth compared with 0, 2 births compared with 1, and 3 births compared with 2). This reparameterization allows for a focus on the effect of each additional birth on cancer risk. The RRs were assumed to be the same regardless of birth number, and the presented RRs are therefore RRs for each additional birth. To allow for a different short-term and long-term effect of pregnancy, RRs were allowed to vary according to time since birth (<10 years, ≥10 years) for parous women. In the presentation of the model we focused on the parameters related to the long-term effect of pregnancy. We furthermore allowed RRs to be different for childbirths at younger (<30 years) and older maternal age (≥30 years) to focus on early age pregnancies which have previously been associated with long-term reduced risk of breast cancer[3,4]. The previously used method (4) was extended to include pregnancy duration. In the previous approach the effect of each birth was stratified according to time since birth and age at childbirth, but in this extended approach it was further stratified by pregnancy duration. Thus, RRs were allowed to vary by duration of the pregnancy in weeks, by the following categories: 20–27, 28–29, 30, 31, …, 41, 42–45 weeks, missing duration of pregnancy, duration of pregnancy not reported, extremely early births (<20 weeks), and extremely late births (>45 weeks). The four last categories are further described in Table 1, Supplementary Table 1, and Supplementary Table 2.

In the analysis of pregnancy duration, all parameters described above were included simultaneously. For example, for biparous women whose first birth occurred in early age at week 38 and whose second birth occurred in late age at week 40, their pregnancy history was modeled by four parameters: the short-term and long-term effect of an early age birth at week 38, and the short-term and long-term effect of a late age birth at week 40. Thus, when estimating the long-term effect of the early age pregnancy at week 38, the model also included the short-term effect of an early age pregnancy at week 38, the short-term effect of an late age pregnancy at week 40, and the long-term of an late age pregnancy at week 40.

The analysis of pregnancy duration was based on follow-up time from January 1, 1978 in Denmark, and from January 1, 1967 in Norway, when the respective Medical Birth Registers began recording gestational week of birth. Childbirths registered in civil registrations systems were incorporated in the analyses to adjust for the effects of pregnancies before start of the Medical Birth Registers.

In analysis of the effect of age at childbirth on breast cancer risk, RRs were allowed to vary according to age at delivery in the categories <20, 20–21, 22–23, 24–25, 26–27, 28–29 and ≥30 years. In analyses of the adjustment effect of socioeconomic status, each socioeconomic variable was added as an additional variable.

All analyses were performed using SAS version 9.4 and procedure GENMOD.

**Socioeconomic factors and risk of breast cancer**. Using nationwide registries from Statistics Denmark on educational attainment, employment and disposable household income starting from respectively 1970, 1976, and 1990, we were able to create a time-varying, three factor adjustment for socioeconomic status. The following categories of the three variables for socioeconomic status were used:

Educational attainment: primary schooling; high school; high school with technical or mercantile focus; short basic education; higher education of short duration; higher education of intermediate duration; academic bachelor degree; academic master's degree; and academic doctoral degree or equivalent educational degree.

Employment status: business owner, ten or more employees; business owner, five to nine employees; business owner, one to four employees; business owner, no employees; business owner, unknown number of employees; co-working spouse; executive officer in business, organization or public office; employee in job which necessitates advanced skills; employee in job which necessitates intermediate skills; employee in job which necessitates basic skills; employee, other; employee, unknown position; unemployed for more than 6 months; social security recipient because of disability; in educational program; disability pensioner; pensioner; early retirement pensioner; social security recipient; other; children under the age of 15 years; housewife (only categorized 1976–1990).

Disposable household income: groups of 10%-percentiles according to the 5-year disposable household income distribution.

**Birthweight and maternal risk of breast cancer**. In order to investigate the effect of birthweight relative to gestational age in pregnancies of different duration, we combined data on gestational age and birthweight from the Danish Medical Birth Registry compiled from 1978 and onwards. We defined a birth small for gestational age (SGA) if the birthweight was below the 10th percentiles of births at the given gestational week, in the corresponding 5-year period. We then stratified by weight category (SGA vs. 10–100th percentile of birthweights at same week) and assessed risk of breast cancer after a pregnancy of given duration, grouped into the following lengths of pregnancy: week 20–33, week 34–36, week 37, 38, 39, 40, 41, and week 42 or longer. To estimate RR of breast cancer by both relative birthweight and gestational period, we extended the model so that the pregnancy effect also varied by relative birthweight.

**Threshold model analysis of pregnancy duration and risk of breast cancer**. Our estimates of the effect of an early age pregnancy stratified by the duration of pregnancy (Fig. 2c) suggest that pregnancies lasting 34 gestational weeks are necessary to obtain a long-term reduced risk of breast cancer. To substantiate this conclusion we compared the observed pattern in Fig. 2c with week-specific threshold models, where breast cancer risk reduction is achieved only by pregnancies with a specific minimal duration or longer. The threshold model with the least difference in fit from the observed pattern in Fig. 2c is interpreted as providing the best estimate for the critical length of pregnancy necessary for the long-term breast cancer risk reduction.

The model used in Supplementary Figure 6 is in the following termed $M_{fig2C}$. In this model, the long-term effect of each early age pregnancy with duration of the pregnancy $w$, is modeled as $\beta_w$, with $w$ noting the gestational week categories described in the paper. We compared $M_{fig2C}$ with simple week-specific threshold models ($M_{threshold}(w_0)$) by which a certain threshold of pregnancy duration is associated with a decreased risk of breast cancer. In that model the long-term effect of each early-age pregnancy according to the duration of the pregnancy $w$ is modeled as $\beta \cdot I(w \geq w_0)$, i.e., by one parameter. With regard to the other parameters in the model, the two models are similar, i.e., a total difference of 14 parameters.

We furthermore compared models that allowed for difference in the pregnancy effect according to parity (primiparity, multiparity) and country (Denmark, Norway). All models were compared by the deviance (i.e., the difference in $-2 \cdot$loglikelihood between two models).

Supplementary Figure 6 shows the deviance between $M_{fig2C}$ and the week-specific $M_{Threshold}(w_0)$ models for different threshold values (in gestational weeks), $w_0$, when using the simple week-specific threshold model (Supplementary Fig. 6A), when allowing for difference in effect according to parity (Supplementary Fig. 6B) and when further allowing for difference in effect according to both parity and country (Supplementary Fig. 6C). The best fit were in all three situations found using a threshold value of 34 weeks.

## Data availability

The data that support the findings of this study are archived at governmental institutions in Denmark and Norway, and can be obtained through application to the relevant data agencies. In Denmark, data from the Medical Birth Registry and the Danish Cancer Registry were retrieved from the Health Data Agency (accession No. FSEID-00002894), while information on income, educational attainment, employment status, and disposable household income was retrieved from Statistics Denmark (accession No. 706117). In Norway, data from the Medical Birth Registry of Norway was retrieved from the Norwegian Institute of Public Health (accession No. 06/930-235) and breast cancer data was retrieved from the Cancer Registry of Norway (accession No. 02/16-623.1).

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

## Acknowledgments

The study was supported by The Health Foundation (Helsefonden), A.P. Møller og Hustru Chastine Mc-Kinney Møllers Fond til almene Formaal, The Danish Cancer Society, The Danish Medical Research Council, and the "Små Forsk" Start grant from Research Council Norway and University of Bergen, Norway. The funders had no role in study design, data collection, data analysis, interpretation of results, writing of the report, and in the decision to submit the article for publication.

## Author contributions

A.H. classified register data, performed statistical analysis, contributed to the study design, interpreted the study results, and drafted the manuscript. J.W. contributed to the study design, planned statistical analysis, oversaw the conduct of the statistical analysis, interpreted the study results, and revised the manuscript. N.Ø. contributed to classification of register data, contributed to the study design, interpreted the study results, and revised the manuscript. M.M. conceived the study, contributed to the study design, interpreted the study results, and revised the manuscript. All authors had access to all of the data and take full responsibility for the integrity of the data, the accuracy of the data analysis, the finished article and the decision to submit the article.

## Additional information

**Competing interests:** The authors declare no competing interests.

