## [Peer Review File · Nature Communications]

Reviewers' Comments:

Reviewer #1:

Remarks to the Author:

Dear Professor Melbye and authors:

Thank you for submitting your work entitled "Pregnancy Duration and Breast Cancer Risk". This work is a well-written, clearly presented manuscript highlighting an extremely large collaborative effort between the Norwegian and Danish databases to generate some very insightful data regarding the duration of a pregnancy and the effect on long term protection of parity on breast cancer risk. To my knowledge, it is a very unique body of work for both its size, robustness of the data generated from these large sets and the focus specifically on duration of pregnancy and its effect on breast cancer protection. The implications for this work are well articulated by the authors and include both understanding of how to view the complexities of pregnant and their outcomes for a women's overall breast cancer risk and, very importantly, offer basic mammary gland biology researchers a target window to better understand the effect of pregnancy on protection, as these data will call many of the current hypotheses into question. The statistical analyses appear appropriate, though I am not a biostatistician.

I have the two following comments as minor revisions to recommend for the manuscript:

1. The rationale behind the selection of comparisons as N=number of pregnancies and the HR of 1 being set to N-1 was not completely clear to me. I understand that allows for the additive effect per pregnancy, but why would you not do both - compare all to 0 and then also the n-1 category? Could you offer a more detailed explanation for the assumption that each birth number does not change the RR more than by 1 birth.

2. Would you consider including in your opening/discussion a sentence or two acknowledging the cross over effect of pregnancy? While I understand it is beyond this body of work to look at early post-pregnancy increase in risk and the focus is solely on later time points, including language in the paper that explains this concept as well would enhance the context of the presented data to the overall field of breast cancer and pregnancy interactions.

Overall, this is a well done body of work that is very important to the breast cancer field and is impactful in its findings.

Virginia F. Borges, MD, MMSc reviewer

Reviewer #2:

Remarks to the Author:

Based on two large population cohorts in Denmark and Norway established through record linkage, the authors reported a reduction of breast cancer risk specifically for those mothers with pregnancies lasting 34 weeks or longer.

Comments:

1. The basic concept of the modeling approach used by the authors was described in the previous paper (Wohlfahrt et al 2001), which delineated the models for age of birth and number of parity. It however doesn't include the variable on pregnancy duration, which is the main variable of interests in this manuscript. One can presume that the authors apply a similar modeling approach, however there are several variations of how one can incorporate the additional variable in this model, which would affect the result interpretation. It would have been much more informative if the authors provided details on how the model is constructed to analyze the duration of pregnancy, accounting for age at birth, time since birth and other covariates.

2. Related to the first point - Was other pregnancy related factors accounted for simultaneously in the analysis for pregnancy duration? For example, age of first child birth, and age of the birth of the corresponding birth?

3. In Table 1, the distribution of the latest pregnancy was shown. Was the analysis of pregnancy duration limited to the last pregnancy? This is currently not clear in the manuscript.

IF yes- it is possible that the pregnancy duration of the last pregnancy can be correlated to the pre-existing medical conditions or age of the mother. Part of these can be further dissected from the data. For those factors that cannot be investigated, the potential implications should be discussed.

IF not restricted to the last pregnancy – how was different pregnancy durations of multiple pregnancies of the same women accounted for ? This is again related to the fact that the modeling approach was not specified clearly in the manuscript.

4. Record linkage through population registry to achieve large sample sizes is a strength. However, the limited ability account for potential confounders would be a main limitation. The authors showed the results with and without adjustment for socioeconomical factors, which was helpful. However, many other potential confounders were not accounted for, for example alcohol consumption, other medical conditions. This should be discussed in the manuscript.

5. Minor points:

a. Page 3: the opening statement regarding the breast cancer etiology is unclear is not exactly accurate. There are substantial amount of data about breast cancer etiology on hormonal factors, anthropometric measures, physical activities and other lifestyle factors, genetics/family history, and more. Suggest to revise this opening statement to better reflect current state of knowledge of breast cancer etiology.

Response to Referees

Reviewers' comments:

R0. We thank both reviewers for their insightful comments regarding our manuscript. Our point-by-point responses are written below:

Reviewer #1 (Remarks to the Author):

Thank you for submitting your work entitled "Pregnancy Duration and Breast Cancer Risk". This work is a well-written, clearly presented manuscript highlighting an extremely large collaborative effort between the Norwegian and Danish databases to generate some very insightful data regarding the duration of a pregnancy and the effect on long term protection of parity on breast cancer risk. To my knowledge, it is a very unique body of work for both its size, robustness of the data generated from these large sets and the focus specifically on duration of pregnancy and its effect on breast cancer protection. The implications for this work are well articulated by the authors and include both understanding of how to view the complexities of pregnancy and their outcomes for a women's overall breast cancer risk and, very importantly, offer basic mammary gland biology researchers a target window to better understand the effect of pregnancy on protection, as these data will call many of the current hypotheses into question. The statistical analyses appear appropriate, though I am not a biostatistician.

I have the two following comments as minor revisions to recommend for the manuscript:

1. The rationale behind the selection of comparisons as N=number of pregnancies and the HR of 1 being set to N-1 was not completely clear to me. I understand that allows for the additive effect per pregnancy, but why would you not do both - compare all to 0 and then also the n-1 category? Could you offer a more detailed explanation for the assumption that each birth number does not change the RR more than by 1 birth.

R1. Previous research on breast cancer have focused on the effect of the age at first childbirth on breast cancer risk. However, as we present in Figure S1, using our longitudinal nationwide data we found a protective effect of each additional early age childbirth. The protective effect of each additional childbirth was not statistically different (Fig. S1), which supported focusing on the 'additional pregnancy effect' in the study of the pregnancy duration, instead of age at/duration of the first childbirth, as have been done previously. We therefore opted for investigating the average effect of pregnancy duration on breast cancer risk, whereby the effect of every pregnancy was taken into account. This approach to modelling breast cancer risk by pregnancy duration gave equivalent results when investigating either the additional effect of all early age childbirths or the additional effect of early age childbirth restricted to women who had given birth once, despite the latter analysis' markedly lower statistical power (compare Fig. 2C to Fig. S2).

In addition, to further exemplify our approach, we expanded the method section with a more detailed explanation of the pregnancy modelling (see R3).

2. Would you consider including in your opening/discussion a sentence or two acknowledging the cross over effect of pregnancy? While I understand it is beyond this body of work to look at early post-pregnancy increase in risk and the focus is solely on later time points, including language in the paper that explains this concept as well would enhance the context of the presented data to the overall field of breast cancer and pregnancy interactions.

R2. We agree with reviewer #1 on the need for explicitly recognizing the dual effect of pregnancy on breast cancer risk. We therefore made the following changes (marked in **bold**) in the introduction:

“Previously, full-term pregnancies in early life (< 30 years) have consistently been associated with a long-term reduced risk of breast cancer(3,4). **Conversely, a transient increased breast cancer risk immediately following full-term pregnancies have been observed (X1)**. Induced abortions and other pregnancies of short duration have, on the other hand, been shown not to influence breast cancer risk(5,6). We hypothesized that by investigating pregnancies of intermediate to long duration in early life (including stillbirths, preterm and term livebirths) we could determine the minimal duration of pregnancy associated with **the long-term** reduced risk of breast cancer and thereby potentially point to underlying mechanisms of the protective effect.”

new reference included in the manuscript:

(X1) M. Lambe, C. Hsieh, D. Trichopoulos et al. *Transient increase in the risk of breast cancer after giving birth. New England Journal of Medicine. 1994.*

Overall, this is a well done body of work that is very important to the breast cancer field and is impactful in its findings.

Reviewer #2 (Remarks to the Author):

Based on two large population cohorts in Denmark and Norway established through record linkage, the authors reported a reduction of breast cancer risk specifically for those mothers with pregnancies lasting 34 weeks or longer.

Comments:

1. The basic concept of the modeling approach used by the authors was described in the previous paper (Wohlfahrt et al 2001), which delineated the models for age of birth and number of parity. It however doesn't include the variable on pregnancy duration, which is the main variable of interests in this manuscript. One can presume that the authors apply a similar modeling approach, however there are several variations of how one can incorporate the additional variable in this model, which would affect the result interpretation. It would have been much more informative if the authors provided details on how the model is constructed to analyze the duration of pregnancy, accounting for age at birth, time since birth and other covariates.

R3. We agree that there is a need for further explanation of how duration of pregnancy was included in the referenced model that “only” included age at birth and times since birth. We have now expanded the statistical method section accordingly and exemplified our approach.

“Pregnancy history was modeled by time-dependent variables as described previously(4). Thus, instead of describing history by the total number of childbirths (i.e. RR of cancer in women with 1, 2, 3 or 4 births compared with women with 0 births), pregnancy history was evaluated by the relative risk for women with n births compared with women with $n-1$ births (i.e. RR of cancer for 1 birth compared with 0, 2 births compared with 1, and 3 births compared with 2). This re-parameterization allows for a focus on the effect of each additional birth on cancer risk. The RRs were assumed to be the same regardless of birth number, and the presented RRs are therefore RRs for each additional birth. To allow for a different short-term and long-term effect of pregnancy, RRs were allowed to vary according to time since birth (<10 years, ≥ 10 years) for parous women. In the presentation of the model we focused on the parameters related to the long-term effect of pregnancy. We furthermore allowed RRs to be different for childbirths at younger (< 30 years) and older maternal age (≥ 30 years) to focus on early age pregnancies which have previously been associated with long-term reduced risk of breast cancer(3,4). **The previously used method (4) was extended to include pregnancy duration. In the previous approach the effect of each birth was stratified according to time since birth and age at childbirth, but in this extended approach it was further stratified by pregnancy duration. Thus, RRs were allowed to vary by duration of the pregnancy in weeks, by the following categories: 20-27, 28-29, 30, 31, ... , 41, 42-45 weeks, missing duration of pregnancy, duration of pregnancy not reported, extremely early births (< 20 weeks), and extremely late births (> 45 weeks). The four last categories are further described in Table 1 and Supplementary Table 1.**

In the analysis of pregnancy duration, all parameters described above were included simultaneously. For example, for biparous women whose first birth occurred in early age at week 38 and whose second birth occurred in late age at week 40, their pregnancy history was modelled by four parameters: the short-term and long-term effect of an early age birth at week 38, and the short-term and long-term effect of a late age birth at week 40. Thus, to estimate the long-term effect of the early age pregnancy at week 38, we simultaneously adjusted for the short-term effect of an early age pregnancy at week 38, the short-term effect of an late age pregnancy at week 40, and the long-term of an late age pregnancy at week 40.

The analysis of pregnancy duration was based on follow-up time from January 1, 1978 in Denmark, and from January 1, 1967 in Norway, when the respective Medical Birth Registers began recording gestational week of birth. Childbirths registered in civil registrations systems were incorporated in the analyses to adjust for the effects of pregnancies before start of the Medical Birth Registers.

In supplementary analysis of the effect of age at childbirth on breast cancer risk, RRs were allowed to vary according to age at delivery in the categories <20, 20-21, 22-23, 24-25, 26-27, 28-29 and ≥ 30 years. In analyses of the adjustment effect of socioeconomic status, each socioeconomic variable was added as an additional variable.”

We furthermore added information on the different categories of pregnancy duration mentioned in Table 1 and Table S1 (new supplementary table):

Supplementary Table 1 is added in the supplementary information.

2. Related to the first point - Was other pregnancy related factors accounted for simultaneously in the analysis for pregnancy duration? For example, age of first child birth, and age of the birth of the

corresponding birth?

R4. See expanded method section above in R3.

3. In Table 1, the distribution of the latest pregnancy was shown. Was the analysis of pregnancy duration limited to the last pregnancy? This is currently not clear in the manuscript.

IF yes- it is possible that the pregnancy duration of the last pregnancy can be correlated to the pre-existing medical conditions or age of the mother. Part of these can be further dissected from the data. For those factors that cannot be investigated, the potential implications should be discussed.

IF not restricted to the last pregnancy – how was different pregnancy durations of multiple pregnancies of the same women accounted for? This is again related to the fact that the modeling approach was not specified clearly in the manuscript.

R5. No, as further explained in the expanded method section (see R3), the analysis of pregnancy duration incorporated the effect of the duration of all pregnancies. To highlight this and the underlying data used for our analysis, we have now included a supplementary table (S1) which shows the distribution of person-years and breast cancer events by first to fifth pregnancy duration.

Supplementary Table 1 is added in the supplementary information.

4. Record linkage through population registry to achieve large sample sizes is a strength. However, the limited ability account for potential confounders would be a main limitation. The authors showed the results with and without adjustment for socioeconomic factors, which was helpful. However, many other potential confounders were not accounted for, for example alcohol consumption, other medical conditions. This should be discussed in the manuscript.

R6. We agree with the reviewer on the need for the discussion of the effect of alcohol consumption, which could be a potential confounder, as it could be associated with both pregnancy duration and breast cancer risk. We have therefore added the following section on the topic to the manuscript:

“High levels of alcohol consumption are found to be associated with an increased risk of breast cancer(X2,X3) and there are reports of an association between heavy alcohol consumption and preterm birth (X4,X5), why alcohol consumption could be a potential factor in the association between pregnancy duration and breast cancer risk. There are however large differences in alcohol consumption between Denmark and Norway, with studies of drinking patterns finding that this is the case both with regards to drinking frequency and volume(X6,X7), with differences in alcohol consumption being especially pronounced during pregnancy(X4,X8). Given the marked differences in alcohol consumption between Denmark and Norway, and the identical findings on the association between pregnancy duration and breast cancer risk, we find it unlikely that alcohol consumption serves as a major confounding factor for the association between pregnancy duration and maternal breast cancer risk.”

new references included in the manuscript:

- (X2) Hamajima N, Hirose K, Tajima K, Rohan T, Calle EE, Heath CW, et al. Alcohol, tobacco and breast cancer--collaborative reanalysis of individual data from 53 epidemiological studies, including 58,515 women with breast cancer and 95,067 women without the disease. *Br J Cancer*. 2002 Nov 18;87(11):1234–45.
- (X3) Tjønneland A, Christensen J, Olsen A, Stripp C, Thomsen BL, Overvad K, et al. Alcohol intake and breast cancer risk: the European Prospective Investigation into Cancer and Nutrition (EPIC). *Cancer Causes Control*. 2007 May 15;18(4):361–73.
- (X4) Albertsen K, Andersen A-MN, Olsen J, Grønbaek M. Alcohol Consumption during Pregnancy and the Risk of Preterm Delivery. *Am J Epidemiol*. 2004 Jan 15;159(2):155–61.
- (X5) Nykjaer C, Alwan NA, Greenwood DC, Simpson NAB, Hay AWM, White KLM, et al. Maternal alcohol intake prior to and during pregnancy and risk of adverse birth outcomes: evidence from a British cohort. *J Epidemiol Community Health*. 2014 Jun;68(6):542–9.
- (X6) Popova S, Lange S, Probst C, Gmel G, Rehm J. Estimation of national, regional, and global prevalence of alcohol use during pregnancy and fetal alcohol syndrome: a systematic review and meta-analysis. *Lancet Glob Heal*. 2017;5(3):e290–9.
- (X7) Wilsnack RW, Wilsnack SC, Kristjanson AF, Vogeltanz-Holm ND, Gmel G. Gender and alcohol consumption: patterns from the multinational GENACIS project. *Addiction*. 2009 Sep;104(9):1487–500.
- (X8) Alvik A, Heyerdahl S, Haldorsen T, Lindemann R. Alcohol use before and during pregnancy: a population-based study. *Acta Obstet Gynecol Scand*. 2006 Jan;85(11):1292–8.

With regards to other medical conditions as potential confounders, we find that we cover the most important confounders that are currently understood to be associated with both pregnancy duration and breast cancer risk. Further, we find support for this conclusion in the similarity of estimates for all women in Fig. 2C (among who a proportion of the nulliparous cohort-members are sub –or infertile) and the estimates for only parous women in Fig. S2.

5. Minor points:

a. Page 3: the opening statement regarding the breast cancer etiology is unclear is not exactly accurate. There are substantial amount of data about breast cancer etiology on hormonal factors, anthropometric measures, physical activities and other lifestyle factors, genetics/family history, and more. Suggest to revise this opening statement to better reflect current state of knowledge of breast cancer etiology.

R6. We agree with the reviewer in that there are well-known, well-described risk factors for breast cancer. However, even with full information on these risk factors, many, if not most, breast cancer cases cannot be predicted prior to diagnosis. The etiology of breast cancer is therefore not clear cut. We therefore suggest the following wording in the opening statement:

“Breast cancer is the most common malignant cancer in women and a major cause of disease burden worldwide(1). **Both** the number and timing of a woman’s childbirths have long been known to influence her breast cancer risk(2), **but how these factors influence breast cancer etiology is not well understood.**”

Reviewers' Comments:

Reviewer #1:

Remarks to the Author:

The authors have been very responsive to both reviewers comments and the manuscript improved in clarity and ability for clear understanding of the methods and results through these edits. It is an important body of work and I believe it should move forward as written.

Response to referees letter

REVIEWERS' COMMENTS:

Reviewer #1 (Remarks to the Author):

The authors have been very responsive to both reviewers comments and the manuscript improved in clarity and ability for clear understanding of the methods and results through these edits. It is an important body of work and I believe it should move forward as written.

R0. We are happy with the response of the reviewer and have made no further changes as the result of this. However, we have changed the manuscript and supplement accordingly with the editorial formatting requests, in addition to a revised acknowledgements statement and the correction of one typo.